# Weakly-supervised High-resolution Segmentation of Mammography Images for Breast Cancer Diagnosis

**Kangning Liu**[1]                      KANGNING.LIU@NYU.EDU

**Yiqiu Shen**[1]                        YS1001@NYU.EDU

**Nan Wu**[1]                         NAN.WU@NYU.EDU

**Jakub Chłędowski**[4]             JAKUB.CHLEDOWSKI@GMAIL.COM

**Carlos Fernandez-Granda**[*1,2]         CFGRANDA@CIMS.NYU.EDU

**Krzysztof J. Geras**[*3,1]                K.J.GERAS@NYU.EDU

[1] *NYU Center for Data Science*

[2] *Courant Institute of Mathematical Sciences, NYU*

[3] *NYU Grossman School of Medicine*

[4] *Jagiellonian University*

## Abstract

In the last few years, deep learning classifiers have shown promising results in image-based medical diagnosis. However, interpreting the outputs of these models remains a challenge. In cancer diagnosis, interpretability can be achieved by localizing the region of the input image responsible for the output, i.e. the location of a lesion. Alternatively, segmentation or detection models can be trained with pixel-wise annotations indicating the locations of malignant lesions. Unfortunately, acquiring such labels is labor-intensive and requires medical expertise. To overcome this difficulty, weakly-supervised localization can be utilized. These methods allow neural network classifiers to output saliency maps highlighting the regions of the input most relevant to the classification task (e.g. malignant lesions in mammograms) using only image-level labels (e.g. whether the patient has cancer or not) during training. When applied to high-resolution images, existing methods produce low-resolution saliency maps. This is problematic in applications in which suspicious lesions are small in relation to the image size. In this work, we introduce a novel neural network architecture to perform weakly-supervised segmentation of high-resolution images. The proposed model selects regions of interest via coarse-level localization, and then performs fine-grained segmentation of those regions. We apply this model to breast cancer diagnosis with screening mammography, and validate it on a large clinically-realistic dataset. Measured by Dice similarity score, our approach outperforms existing methods by a large margin in terms of localization performance of benign and malignant lesions, relatively improving the performance by 39.6% and 20.0%, respectively. Code and the weights of some of the models are available at https://github.com/nyukat/GLAM.

**Keywords:** weakly supervised learning, high-resolution medical images, breast cancer screening

## 1. Introduction

Convolutional neural networks (CNNs) have revolutionized medical image analysis (Bekkers et al., 2018; Ching et al., 2018; Topol, 2019; Prevedello et al., 2019; Lutnick et al., 2019; Karimi et al., 2020; Gaonkar et al., 2021). These networks achieve impressive results, but their outputs are often difficult to interpret, which is problematic for clinical decision making (Yao et al., 2018). Designing

---

* Contributed equally

explainable classification models is a challenge. In some applications, such as cancer diagnosis, interpretability can be achieved by localizing the regions of the input image that determine the output of the model (Shen et al., 2021). Alternatively, detection and segmentation networks, such as U-Net (Ronneberger et al., 2015) and Faster R-CNN (Ren et al., 2016) can be trained with annotations indicating regions relevant to diagnosis. Unfortunately, acquiring such annotations is labor-intensive, and requires medical expertise. Moreover, learning under such supervision might bias the network to ignore lesions occult to radiologists, which a neural network can still identify.

Given the above obstacle, weakly-supervised localization (WSL) has recently become an area of active research (Diba et al., 2017; Singh and Lee, 2017; Zhang et al., 2018a,b; Cui et al., 2019). These approaches aim to identify image regions relevant to classification utilizing only image-level labels during training, based upon the observation that feature maps in the final convolutional layers of CNNs reveal the most influential regions of the input image (Oquab et al., 2015; Zhou et al., 2016). These methods are usually designed for natural images and applying them to medical images is challenging due to their unique characteristics. For example, mammography images have a much higher resolution ($\sim 10^7$ pixels) than natural images ($\sim 10^5$ pixels) in most benchmark datasets, such as ImageNet (Deng et al., 2009). Because of this, when applied to medical images, CNNs often aggressively downsample the input image (Shen et al., 2019, 2021) to accommodate GPU memory constraints, making the resulting localization too coarse. This is a crucial limitation for many medical diagnosis tasks, where regions of interest (ROIs) are often small (e.g. $\leq 1\%$ pixels).

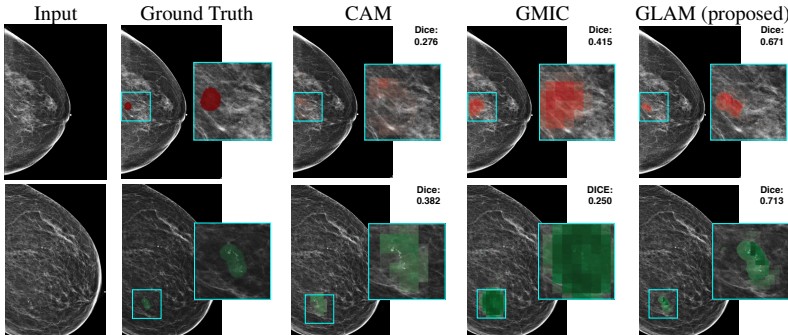

Figure 1: Comparison of saliency maps generated by CAM (Zhou et al., 2016), GMIC (Shen et al., 2021), and the proposed method on a mammography image containing a malignant lesion (first row, red) and a benign lesion (second row, green). Both CAM and GMIC produce coarse saliency maps that fail to localize the lesion accurately. The proposed method generates a high-resolution saliency map that precisely localizes the lesions.

In this work, we propose GLAM (Global-Local Activation Maps), a novel framework to generate fine-grained segmentation using only image-level labels. The proposed model processes high resolution medical images in a memory-efficient way. The main idea behind GLAM is to select informative regions (patches) that may contain ROIs via coarse-level localization and then to perform segmentation on selected patches rather than the entire image in a weakly supervised manner. We train and evaluate GLAM on a dataset containing more than one million mammography images. We demonstrate that the model outperforms existing baselines in segmentation of both benign and malignant lesions, improving the Dice similarity score relatively by 39.6% and 20%, respectively,

while preserving classification accuracy. To achieve that, GLAM produces fine-grained saliency maps with a 300 times higher resolution than previous works (Shen et al., 2021) ($736 \times 480$ pixels for $2944 \times 1920$ pixels input images). In Figure 1, we illustrate how the saliency maps generated by GLAM enable high-resolution segmentation of lesions relevant to breast cancer diagnosis.

## 2. Background

WSL is the task of learning to locate ROIs (i.e. the *objects*) in an image when only image-level labels are available during training. WSL methods are usually based on CNNs that produce saliency maps encoding the location of ROIs. To train the whole system using only image-level labels, the saliency maps are collapsed to predictions indicating the presence of each class using a pooling function. Once the CNN is trained, the saliency map can be used for localization (Oquab et al., 2015; Zhou et al., 2016). WSL has been applied in a wide range of medical-imaging applications, including the detection of lung disease in chest X-ray images (Wang et al., 2017; Yao et al., 2018; Tang et al., 2018; Ma et al., 2019; Liu et al., 2019; Guan et al., 2018), diagnosis of injuries from pelvic X-ray images (Wang et al., 2019), brain lesion segmentation (Wu et al., 2019a), breast MRI analysis (Luo et al., 2019), cancer detection in lung CT images (Feng et al., 2017; Schlemper et al., 2018), and scan-plane detection in ultrasound (Schlemper et al., 2018; Baumgartner et al., 2017).

The majority of these works focus on images that have relatively low resolution, that is, $512 \times 512$ pixels or less. Only a few works have considered higher resolution images, which are standard in some imaging procedures such as screening mammography (Shen et al., 2019, 2021).

In this work we focus on the diagnosis of breast cancer from screening mammography images. Breast cancer is the second leading cause of cancer-related deaths among women (Bray et al., 2018) and screening mammography is the main tool for its early detection (Marmot et al., 2013). CNN classifiers have shown promise in diagnosis from mammograms (Zhu et al., 2017; Kim et al., 2018; Ribli et al., 2018; Wu et al., 2019b; Geras et al., 2019; McKinney et al., 2020; Shen et al., 2019, 2021). Accurate localization of suspicious lesions is crucial to aid clinicians in interpreting model outputs, and can provide guidance for future diagnostic procedures. However, existing methods that explain their predictions, e.g. Shen et al. (2019, 2021), offer only coarse localization. GLAM is inspired by recent works (Yao et al., 2018; Shen et al., 2019, 2021; Shamout et al., 2020), which improve classification accuracy by processing image patches selected from coarse saliency maps. The main innovation of GLAM with respect to these works is that it generates a a high-resolution saliency map from the selected patches, which significantly improves lesion segmentation accuracy.

## 3. Proposed Approach

Our goal is to generate fine-grained saliency maps that localize objects of interest in high-resolution images using only image-level labels during training. We start this section by describing the inference pipeline of our approach in Section 3.1. We then describe each module in detail in Section 3.2. Finally, we explain the training strategy in Section 3.3.

### 3.1. Inference pipeline

As illustrated in Figure 2, during inference, our system processes an input $\mathbf{x} \in \mathbb{R}^{H,W}$ as follows:

1. The image $\mathbf{x}$ is fed into the *global module*, a memory-efficient CNN denoted by $f_g$, to produce an image-level coarse saliency map $\mathbf{S}_g$ and an image-level class prediction $\hat{y}_g$.

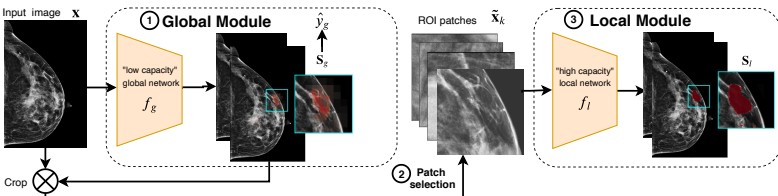

Figure 2: Inference pipeline of GLAM. 1) The global network $f_g$ is applied to the whole image $\mathbf{x}$ to obtain a coarse image-level segmentation map $\mathbf{S}_g$. 2) Based on this coarse-level segmentation, several patches are extracted from the input image. 3) The local network $f_l$ processes these patches to generate a high-resolution saliency map $\mathbf{S}_l$.

2. We select $M$ patches from $\mathbf{x}$ based on $\mathbf{S}_g$. To do that, we greedily select the patches for which the sum of the entries in $\mathbf{S}_g$ is the largest (see Algorithm 1 for a detailed description).

3. We feed the selected patches $\tilde{\mathbf{x}}_1, \ldots, \tilde{\mathbf{x}}_M$ to the *local module* $f_l$, another CNN which produces a fine-grained saliency map associated with each patch. We then remap the patch-level saliency maps back to their location in the original input image. We denote the saliency map obtained through this procedure by $\mathbf{S}_l$.

GLAM produces an image-level saliency map $\mathbf{S}_g$ and a fine-grained multi-patch saliency map $\mathbf{S}_l$. These maps are aggregated through averaging to produce the final saliency map $\mathbf{S}_c = (\mathbf{S}_g + \mathbf{S}_l)/2$. In addition, GLAM generates a classification output, which is produced by the global module (as this yields the best classification accuracy).

### 3.2. Module parameterization

**Global module**    The architecture of $f_g$ is based on the design of Shen et al. (2019, 2021), which is similar to ResNet (He et al., 2016) with a reduced number of channels for memory efficiency. The main difference between $f_g$ and the global module of Shen et al. (2019, 2021) is that we combine saliency maps at different scales to generate the global saliency map, inspired by Sedai et al. (2018) who observed that using convolutional feature maps extracted only from the last layer of a CNN may be suboptimal in localization of small objects. We generate saliency maps at different scales ($\mathbf{S}_0$, $\mathbf{S}_1$ and $\mathbf{S}_2$) using a pyramidal hierarchy of feature maps (see Figure 6 in Appendix B). Our image-level saliency map $\mathbf{S}_g$ is obtained by averaging them. For each $\mathbf{S}_n$ ($n \in \{0, 1, 2\}$), we obtain a classification prediction $\tilde{y}_n$ associated with $\mathbf{S}_n$ using $\mathrm{top}\,t\%$ pooling (see Appendix B), where $t$ is a hyperparameter. The image-level classification prediction $\hat{y}_g$ is calculated by averaging $\tilde{y}_0$, $\tilde{y}_1$, and $\tilde{y}_2$. Additionally, we output a representation vector $z_g$ to feed it into a fusion module (described below) to enable joint training with the local module. See Appendix B for more details.

**Local module**    Our local module is based on ResNet-34 with a reduced stride in the residual blocks to maintain a higher resolution. We replace the global average pooling and the last fully connected layer by a $1 \times 1$ convolution layer followed by a sigmoid non-linearity. The local module is applied to each of the selected patches $\tilde{\mathbf{x}}_k$ ($k \in 1, \ldots, K$) to extract a patch-level saliency map $\mathbf{A}_k$. As we only have image-level labels, we need to train the patch-level network without patch-level labels. To address this challenge, we combine insights from weakly-supervised localization and multiple-instance learning to train the patch-level saliency maps hierarchically. In multiple

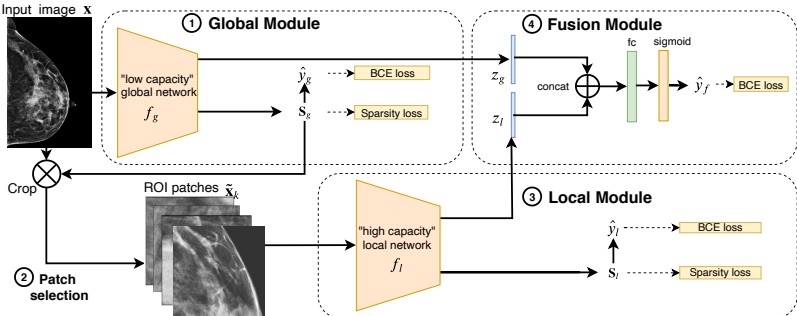

Figure 3: Proposed training strategy. 1) Train the global module and select the best segmentation model. 2) Freeze the global module and use it to extract input patches for the local module. 3) Train the local module on the selected patches. 4) Joint training with the fusion module. We use BCE loss and sparsity loss to train the system.

instance learning (Maron and Lozano-Pérez, 1998) the labels are associated with bags of instances (the label is negative if all instances in a bag are negative, and positive otherwise). In our case each instance is a patch, and the patches from an image form a bag. We use a patch aggregation function $f_\text{p}$ to combine the information across all patches and form an image-level prediction $\hat{y}_l$. Formally, we have $\hat{y}_l = f_\text{p}(\mathbf{A_1}, \ldots, \mathbf{A_k})$. We propose two different patch aggregation functions.

- *Concatenation-based aggregation*: We concatenate the saliency maps spatially and apply pooling function $f_\text{agg}$ (*i.e.* top $t\%$ pooling). The prediction is thus given by $\hat{y}_l = f_\text{agg}(\text{concat}(\mathbf{A}_1, \ldots, \mathbf{A}_K))$.
- *Attention-based aggregation:* top $t\%$ pooling is applied to $\mathbf{A}_k$ to obtain a patch-level prediction $\hat{y}_k$. Additionally, we output a representation vector $z_k$ for each patch, which will be aggregated to $z_l$ and fed into the fusion module. We use the Gated Attention Mechanism ((Ilse et al., 2018)) to combine the prediction and the representation vectors using attention weights $\alpha_i \in [0, 1]$. The prediction is given by $\hat{y}_l = \sum_{i=1}^K \alpha_i \hat{y}_i$, and the representation vector by $z_l = \sum_{i=1}^K \alpha_i z_i$.

Section 3.3 explains when we use each aggregation method. Refer to Appendix C for more details.

**Fusion module**   In order to jointly optimize the parameters of the global and local modules, we incorporate a *fusion module* consisting of one fully connected layer that combines the representation vectors from the global ($z_g$) and local modules ($z_l$) to produce a fusion prediction $\hat{y}_f$. Formally, $\hat{y}_f = \text{sigmoid}(w_f[z_g, z_l]^T)$, where $w_f$ is a vector of learnable parameters.

### 3.3. Training strategy

The training strategy that achieves the best experimental performance for GLAM is sequential. We first train the global module, then we train the local module and finally we train both together. This makes sense because in order to train the local module $f_l$ effectively, we need to select meaningful input patches. Since the selection relies on the global saliency map, this requires pretraining the global module. Our training strategy is as follows (see also Figure 3).

1. Train the global module with the loss function $L_g = \sum_{n\in\{0,1,2\}}(\text{BCE}(y, \tilde{y}_n) + \lambda \sum_{(i,j)} |\mathbf{S}_n(i,j)|)$ using the whole training set. Here, $\text{BCE}(y, \tilde{y}_n)$ is the binary cross-entropy loss and $\sum_{(i,j)} |\mathbf{S}_n(i,j)|$ is an $\ell_1$ regularization enforcing sparsity of the saliency map. The hyperparameter $\lambda$ balances the two terms. The model is selected based on segmentation performance on the validation set.

2. Create training data for the local module. To produce negative examples, we select $K$ patches randomly from images with negative labels. The positive examples are generated by selecting $K$ patches from images with positive labels using Algorithm 1 applied to the saliency maps produced by the pretrained global module.

3. Train the local module minimizing the cost function $L_l = \text{BCE}(y, \hat{y}_l) + \lambda \sum_{(i,j)} |\mathbf{S}_l(i,j)|$. Here we apply the concatenation-based aggregation described in Section 3.2 because it yields the best results experimentally and speeds up training.

4. Train the global and local modules jointly with the fusion module. Here we use the attention-based aggregation in the local module. The fusion module outputs a fusion prediction $\hat{y}_f$. The loss function for joint training is $L_t = L_g + L_l + L_f$, where $L_f = \text{BCE}(y, \hat{y}_f)$.

Note that the number of patches selected as input to the local module during training ($K$) and inference ($M$) do not need to be the same. In fact, we find empirically that it is beneficial to use $K$ as large as possible (within the constraints imposed by GPU memory), which is consistent with Shen et al. (2021). However, choosing a large $M$ increases the false positive rate. We used $K = 6$ and $M = 1$, based on experiments reported in Appendix G.4 and G.5.

## 4. Experiments

### 4.1. Dataset and evaluation metrics

We trained and evaluated the proposed model on the NYU Breast Cancer Screening Dataset v1.0 (Wu et al., 2019c) that includes 229,426 exams (1,001,093 images) from 141,472 patients. Each exam contains at least four images with a resolution of $2944 \times 1920$ pixels, corresponding to the four standard views used in screening mammography: R-CC (right craniocaudal), L-CC (left craniocaudal), R-MLO (right mediolateral oblique) and L-MLO (left mediolateral oblique). The dataset is divided into disjoint training (186,816), validation (28,462) and test (14,148) sets, ensuring that each patient only belongs to one of the sets. Each breast has two binary labels indicating whether malignant or benign lesions are present. A subset of the images with lesions have pixel-level annotations provided by radiologists, which indicate the position of the lesions. Note that the dataset (458,852 breasts) contains more exams without lesions (452,311 compared to 5,556 with benign lesions, and 985 with malignant lesions). To account for this imbalance when training our models, at each epoch we use all exams with lesions and an equal number of randomly-sampled exams without lesions.

To measure classification performance, we report the area under the ROC curve (AUC) for identifying breasts with both malignant and benign lesions. To evaluate localization ability, we use the Dice similarity coefficient and pixel average precision (PxAP) (Choe et al., 2020). PxAP is the average of the area under the precision-recall curve for each pixel. The threshold to compute the precision and recall is either chosen for each image (image-level PxAP) or fixed for the whole test set (dataset-level PxAP). These metrics are described in more detail in Appendix E.

### 4.2. Comparison to baselines

We compare GLAM to two baselines: GMIC (Shen et al., 2021), a WSOL method specifically designed for high-resolution medical images, and CAM (Zhou et al., 2016), one of the most popular WSOL methods for natural images. We use the same backbone architecture for the two baselines and the global module of GLAM. The same hyperparameter tuning is applied to all models to ensure

a fair comparison, as described in Appendix F. The models are selected based on segmentation performance, which is not necessarily equivalent to selection based on classification performance (see Appendix D for further discussion). We also compare to a U-Net (Ronneberger et al., 2015) trained with strong supervision using the pixel-level annotations. This provides an upper bound for the segmentation performance of the WSL methods. In Table 1, we report the performance of GLAM and the baselines. GLAM outperforms both GMIC and CAM in all segmentation evaluation metrics by a large margin, while achieving very similar classification accuracy. We observe a performance gap between GLAM and the strongly supervised U-Net model trained with ground-truth segmentation annotations, which indicates that there is room for further improvement.

Table 1: Segmentation performance of our method (GLAM) and several baselines evaluated in terms of Dice (mean and standard deviation over the test set), image-level PxAP and dataset-level PxAP for malignant and benign lesions. We also report the classification AUC achieved by each model. GLAM outperforms the baselines, while achieving very similar classification accuracy. The performance of a model (U-Net) trained with segmentation annotations is also included for comparison.

| | Dice | | image-level PxAP | | dataset-level PxAP | | classification AUC | |
|---|---|---|---|---|---|---|---|---|
| | Malignant | Benign | Malignant | Benign | Malignant | Benign | Malignant | Benign |
| GLAM (proposed) | **0.390 ± 0.253** | **0.335 ± 0.259** | **0.461 ± 0.296** | **0.396 ± 0.315** | **0.341** | **0.215** | 0.882 | 0.770 |
| GMIC | 0.325 ± 0.231 | 0.240 ± 0.175 | 0.396 ± 0.275 | 0.283 ± 0.244 | 0.295 | 0.112 | 0.886 | 0.780 |
| CAM | 0.250 ± 0.221 | 0.207 ± 0.180 | 0.279 ± 0.240 | 0.222 ± 0.210 | 0.226 | 0.084 | 0.894 | 0.770 |
| U-Net (fully supervised) | 0.504 ± 0.283 | 0.412 ± 0.316 | 0.589 ± 0.329 | 0.498 ± 0.357 | 0.452 | 0.265 | - | - |

### 4.3. Ablation study

We analyze different design choices in GLAM through an extensive ablation study. Due to space constraints, here we only discuss the advantages of training the global and local modules jointly, and the segmentation properties of the global and local saliency maps. We defer results on the following choices to the appendices: design of the global module (Appendix G.1), selection of training data for the local module (Appendix G.2), design of the local module (Appendix G.3), number of input patches used by the local module during training (Appendix G.4) and inference (Appendix G.5), the fusion module (Appendix G.6), and hyper-parameter selection: (Appendix G.7, G.8 and G.9).

**Joint training of local and global modules.** Here we compare the GLAM saliency map $\mathbf{S}_{c-joint}$ obtained via joint training as described in Section 3.3 with (1) the global saliency map $\mathbf{S}_g$ from the global module pretrained in isolation, (2) the local saliency map $\mathbf{S}_l$ for the local module trained using patches selected using a frozen global module, (3) a saliency map $\mathbf{S}_{c-sep}$ obtained by averaging $\mathbf{S}_g$ and $\mathbf{S}_l$. The results reported in Table 2 show that $\mathbf{S}_{c-sep}$ is superior to $\mathbf{S}_g$ and $\mathbf{S}_l$, but joint training achieves better performance across all metrics.

**Segmentation properties of the global and local saliency maps** In Figure 4, we plot the Dice scores of the global ($\mathbf{S}_g$) and local ($\mathbf{S}_l$) saliency maps generated by GLAM for 400 randomly selected examples from the validation set. Each saliency map has different strengths and weaknesses. $\mathbf{S}_l$ fails completely for a subset of examples, this is because the patch-selection procedure did not select the correct patches (lower row in Figure 5). On the remaining examples, $\mathbf{S}_l$ tends to outperform $\mathbf{S}_g$ because it has a much higher resolution. However, in some cases it underperforms

Table 2: Segmentation performance of the GLAM saliency map $\mathbf{S}_{c-joint}$, the global ($\mathbf{S}_g$) and local saliency ($\mathbf{S}_l$) maps trained separately, and the average of $\mathbf{S}_g$ and $\mathbf{S}_l$ with ($\mathbf{S}_{c-joint}$) and without ($\mathbf{S}_{c-sep}$) joint training. Averaging helps, joint training further improves.

| | Dice | | image-level PxAP | | dataset-level PxAP | |
|---|---|---|---|---|---|---|
| | Malignant | Benign | Malignant | Benign | Malignant | Benign |
| $\mathbf{S}_g$ | $0.325 \pm 0.239$ | $0.261 \pm 0.185$ | $0.363 \pm 0.276$ | $0.302 \pm 0.266$ | 0.324 | 0.140 |
| $\mathbf{S}_l$ | $0.343 \pm 0.283$ | $0.297 \pm 0.287$ | $0.444 \pm 0.337$ | $0.337 \pm 0.310$ | 0.207 | 0.126 |
| $\mathbf{S}_{c-sep}$ | $0.375 \pm 0.264$ | $0.318 \pm 0.243$ | $0.449 \pm 0.319$ | $0.382 \pm 0.317$ | 0.340 | 0.191 |
| $\mathbf{S}_{c-joint}$ | $\mathbf{0.390 \pm 0.253}$ | $\mathbf{0.335 \pm 0.259}$ | $\mathbf{0.461 \pm 0.296}$ | $\mathbf{0.396 \pm 0.315}$ | **0.341** | **0.215** |

$\mathbf{S}_g$, often in cases where the ground-truth segmentation is larger than the size of the patch (upper row in Figure 5). Averaging $\mathbf{S}_l$ and $\mathbf{S}_g$ (our strategy of choice in GLAM) achieves high-resolution segmentation, while hedging against the failure of the local module.

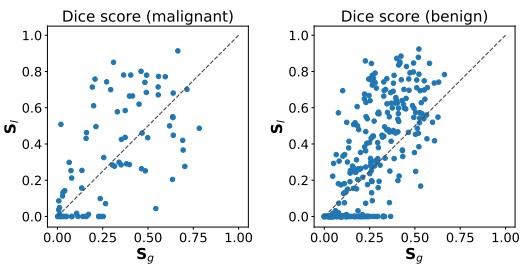

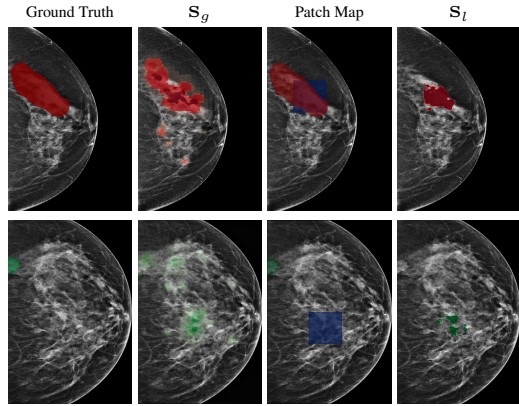

Figure 4: Scatter plot of Dice score of the global $\mathbf{S}_g$ and local $\mathbf{S}_l$ modules for 400 validation examples. $\mathbf{S}_l$ outperforms $\mathbf{S}_g$ for most small lesions, but may miss some larger lesions and fails entirely if the wrong patches are selected as input.

Figure 5: Two failure cases of the local module. (Top) The lesion is larger than the input patch, so $\mathbf{S}_l$ only captures it partially. (Bottom) The input patches to $\mathbf{S}_l$ (in blue) do not cover the lesion.

## 5. Conclusion

In this work, we propose a novel framework to perform weakly-supervised segmentation of images with very high resolution, which outperforms existing methods. Our results suggest that the general principle underlying GLAM (hierarchical selection of saliency maps) could be effective in other applications involving high-resolution images and videos. Another interesting question for future research is how to extend this principle to settings where high-resolution segmentation is desired, but (in contrast to breast cancer) the regions of interest do not tend to be small.

## Acknowledgments

The authors would like to thank Mario Videna, Abdul Khaja and Michael Costantino for supporting our computing environment and Catriona C. Geras for proofreading the paper. We also gratefully acknowledge the support of Nvidia Corporation with the donation of some of the GPUs used in this research. C.F. was partially supported by NSF DMS grant 2009752. K.L. was partially supported by NIH grant R01 LM013316 and NSF NRT grant HDR-1922658. This work was supported in part by grants from the National Institutes of Health (P41EB017183, R21CA225175), the National Science Foundation (1922658) and the Gordon and Betty Moore Foundation (9683).

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

## Appendix A. Patch selection algorithm

---

**Algorithm 1** Patch selection algorithm: $\mathbf{x}$ denotes the input image, $\mathbf{S}$ denotes the saliency map that the patch selection method is based on (in our case of interest, this is the saliency map $\mathbf{S_g}$ from the global module). $\tilde{\mathbf{x}}_k$ denotes the selected patches cropped from the original image. $K$ is the predefined number of selected patches.

---

Require: $\mathbf{x} \in \mathbb{R}^{H,W}, \mathbf{S} \in \mathbb{R}^{h,w,|\mathbb{C}|}, K$      7: $f_c(l, \hat{\mathbf{S}}) = \sum_{(i,j)\in l} \hat{\mathbf{S}}[i,j]$

Ensure: $O = \left\{ \tilde{\mathbf{x}}_k \mid \tilde{\mathbf{x}}_k \in \mathbb{R}^{h_c,w_c} \right\}$      8: for each $1, 2, \ldots, K$ do

1: $O = \emptyset$                                                     9:     $l^* = \text{argmax}_l\, f_c(l, \hat{\mathbf{S}})$

2: for each class $c \in \mathbb{C}$ do                  10:     $L = $ position of $l^*$ in $\mathbf{x}$

3:     $\tilde{\mathbf{S}}^c = \min - \max - \text{normalization}\ (\mathbf{S}^c)$      11:     $O = O \cup \{L\}$

4: end for                                          12:     $\hat{\mathbf{S}}[i,j] = 0, \forall (i,j) \in l^*$

5: $\hat{\mathbf{S}} = \sum_{c\in\mathbb{C}} \tilde{\mathbf{S}}^c$                       13: end for

6: $l$ denotes an arbitrary $h_c\frac{h}{H} \times w_c\frac{w}{W}$      14: return $O$

rectangular patch on $\hat{\mathbf{S}}$

---

We perform patch selection using the same greedy algorithm (Algorithm 1) as in GMIC (Shen et al., 2021). In each iteration, we select the rectangular region in the coarse saliency map generated by the global module, which has the largest average intensity (see line 7 of Algorithm 1). Then the region is interpolated so that it maps to the corresponding location on the input image. Line 12 ensures that the extracted ROI patches do not significantly overlap with each other.

It is worth noting that we do not assume that all patches extracted from a positive example contain tumors: only some of the $K$ patches extracted by the global module will. This is accounted for by the patch aggregation function in the local module, which produces a single classification output for the $K$ aggregated patches. A problem arises if the patch-selection algorithm fails to select any patches containing lesions for a positive example. This is why it is beneficial to use a larger number of patches during training (see Appendix H.4). It is worth noting, however, that this situation does not arise often in our experiments. In order to verify this, we used a set of images with

lesions from the validation set, and checked the three first patches extracted by the global module. These patches did not contain any lesions only in 40 out of 300 images with benign lesions (13%) and in 6 out of 75 images with malignant lesions (8%).

## Appendix B. Global module

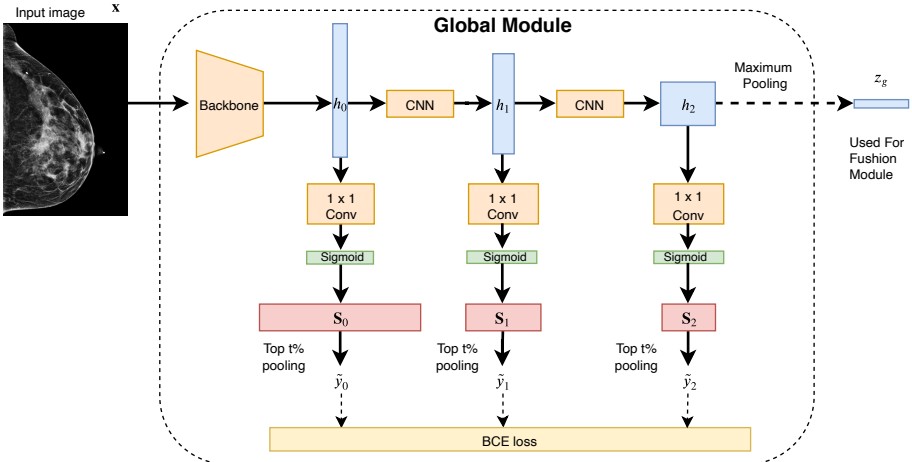

Figure 6: Architecture of the global module. Both high-level ($\mathbf{h}_2 \in \mathbb{R}^{46\times30\times256}$) and low-level feature maps ($\mathbf{h}_0 \in \mathbb{R}^{184\times120\times64}$, $\mathbf{h}_1 \in \mathbb{R}^{92\times60\times128}$) are utilized to obtain multi-scale saliency maps ($\mathbf{S}_0 \in \mathbb{R}^{184\times120\times2}$, $\mathbf{S}_1 \in \mathbb{R}^{92\times60\times2}$ and $\mathbf{S}_2 \in \mathbb{R}^{46\times30\times2}$). For each scale $n \in \{0, 1, 2\}$, we use $\mathrm{top}\,t\%$ pooling to transform saliency maps $\mathbf{S}_n$ into class predictions $\tilde{y}_n$. A combined cross-entropy loss from the three scales is used for training. We compute the global saliency map $\mathbf{S}_g$ by combining the individual saliency maps from three scales ($\mathbf{S}_0$, $\mathbf{S}_1$ and $\mathbf{S}_2$). We apply maximum pooling on the spatial dimension of $h_2 \in \mathbb{R}^{46\times30\times256}$ to obtain the representation vector $z_g \in \mathbb{R}^{256}$, which is fed to the fusion module during joint training.

As shown in Figure 6, the global module provides a multi-scale pyramidal hierarchy of feature maps ($\mathbf{h_0}, \mathbf{h_1}, \mathbf{h_2}$) when processing an input image. The feature map corresponding to the deepest layer ($\mathbf{h_2}$) has low resolution due to downsampling and therefore has limited localization ability. However, it is useful for classification, because it aggregates information from the whole image. Sedai et al. (2018) found that using only the deepest feature maps can negatively influence the localization of small objects. This motivates combining feature maps at several layers (in our case $\mathbf{h_0}, \mathbf{h_1}, \mathbf{h_2}$) to produce saliency maps (Feng et al., 2017; Sedai et al., 2018; Yao et al., 2018). To do this, we use a training loss that includes classification estimates obtained from the three feature maps (see Eq. (1) below).

As shown in Figure 6, given a gray-scale input image of $\mathbf{x} \in R^{2944\times1920}$ pixels, we have feature maps in three scales ($\mathbf{h_0} \in \mathbb{R}^{184\times120\times64}$, $\mathbf{h_1} \in \mathbb{R}^{92\times60\times128}$ and $\mathbf{h_2} \in \mathbb{R}^{46\times30\times256}$). For each feature map $h_n$ of scale $n$, we use $1 \times 1$ convolution followed by a sigmoid function to transform $h_n$ to a saliency map $\mathbf{S_n} \in [0, 1]^{h\times w\times|\mathbb{C}|}$, where $\mathbb{C} = \{\mathrm{malignant}, \mathrm{benign}\}$. Each

element of $\mathbf{S_n}$, $\mathbf{S_n}^c(i,j)$, represents the contribution of a spatial location $(i,j)$ towards classifying the input as class $c \in \mathbb{C}$. We apply $\mathrm{top}\, t\%$ pooling (Shen et al., 2021) as an aggregation function $f_{\mathrm{agg}}(\mathbf{S_n}^c) : \mathbb{R}^{h,w} \mapsto [0,1]$ to transform $\mathbf{S_n}^c$ to the image-level class prediction for class $c$. Formally, $\tilde{y}_n^c = f_{\mathrm{agg}}(\mathbf{S_n}^c) = \frac{1}{|H^+|} \sum_{(i,j) \in H^+} \mathbf{S_n}^c(i,j)$, where $H^+$ denotes the set containing the locations of the top $t\%$ values in $\mathbf{S_n}^c$, where $t$ is a hyperparameter. During training we minimize a sum of the cross-entropy losses between the image-level label $y$ and $\tilde{y}_n$ for the three scales ($n \in \{0,1,2\}$). The training loss function of the global module is the following:

$$L_g = \sum_{n \in \{0,1,2\}} \left( \mathrm{BCE}(y, \tilde{y}_n) + \lambda \sum_{(i,j)} |\mathbf{S}_n(i,j)| \right). \tag{1}$$

Here, $\mathrm{BCE}(y, \tilde{y}_n)$ is the cross-entropy loss, $\sum_{(i,j)} |\mathbf{S}_n(i,j)|$ is a $\ell_1$ regularization term enforcing sparsity of the saliency map (Shen et al., 2021) and $\lambda$ is a hyperparameter to balance the two loss terms.

We obtain the global module prediction $\hat{y}_g$ by taking the average of the class prediction across the levels $\hat{y}_g = (\tilde{y}_0 + \tilde{y}_1 + \tilde{y}_2)/3$. We obtain the global module saliency map $\mathbf{S}_g$ by combining $\mathbf{S}_0$, $\mathbf{S}_1$ and $\mathbf{S}_2$. Note that the resolutions of $\mathbf{S}_0$, $\mathbf{S}_1$ and $\mathbf{S}_2$ are different ($\mathbf{S}_0 \in \mathbb{R}^{184 \times 120 \times 2}$, $\mathbf{S}_1 \in \mathbb{R}^{92 \times 60 \times 2}$ and $\mathbf{S}_2 \in \mathbb{R}^{46 \times 30 \times 2}$). In order to combine them, we upsample $\mathbf{S}_0$ and $\mathbf{S}_1$ to match the resolution of the $\mathbf{S}_2$ using nearest-neighbor interpolation. Then, we compute the global module saliency map as $\mathbf{S}_g = \gamma_0 \mathbf{S}_0 + \gamma_1 \mathbf{S}_1 + \gamma_2 \mathbf{S}_2$. Here $\gamma_0$, $\gamma_1$ and $\gamma_2$ are hyperparameters that can be tuned on the validation set. They should satisfy the conditions: $\gamma_0 + \gamma_1 + \gamma_2 = 1$ and $\gamma_0 \geq 0$, $\gamma_1 \geq 0$, $\gamma_2 \geq 0$. We set $\gamma_0 = 0.2, \gamma_1 = 0.6, \gamma_3 = 0.2$ according to the Dice score on the validation set. We apply maximum pooling on the spatial dimension of $\mathbf{h_2} \in \mathbb{R}^{46 \times 30 \times 256}$ to obtain a representation vector for the global module $z_g \in \mathbb{R}^{256}$, which is fed into the fusion module during joint training of the global and local modules.

Utilizing multi-scale feature maps improves the segmentation performance of $\mathbf{S}_g$. As a result, we obtain a better feature map $\mathbf{S}_c$. This is demonstrated empirically in the ablation study reported in Appendix G.1.

## Appendix C. Local module

We design the local module in order to preserve as much spatial resolution as possible. The resolution of the global saliency map is only $184 \times 120$ pixels. Therefore, the global module backbone is not suitable as a local module backbone. Typical classification networks also do not suit our needs. In our case, the size of the patches that are processed by the local module is $512 \times 512$. For such input resolution, the size of the saliency map produced by ResNet-34 is only $28 \times 28$ pixels. In order to preserve spatial resolution, we build the backbone for the local module by reducing the stride in all ResNet blocks of the ResNet-34 network to 1. When operating on an input patch with size $512 \times 512$ pixels, the resulting network outputs a saliency map with a resolution of $127 \times 127$ pixels. Note that we use ResNet-34 rather than ResNet-18 because it has a larger receptive field, which compensates for the reduced receptive field due to the reduced stride. We provide an ablation study of different choices for the local-module backbone architecture in Appendix G.3.

Architecture of the backbone of the local module is described in the left panel of Figure 7. In the right panel, we show two patch-aggregation strategies. We report an ablation study on the value

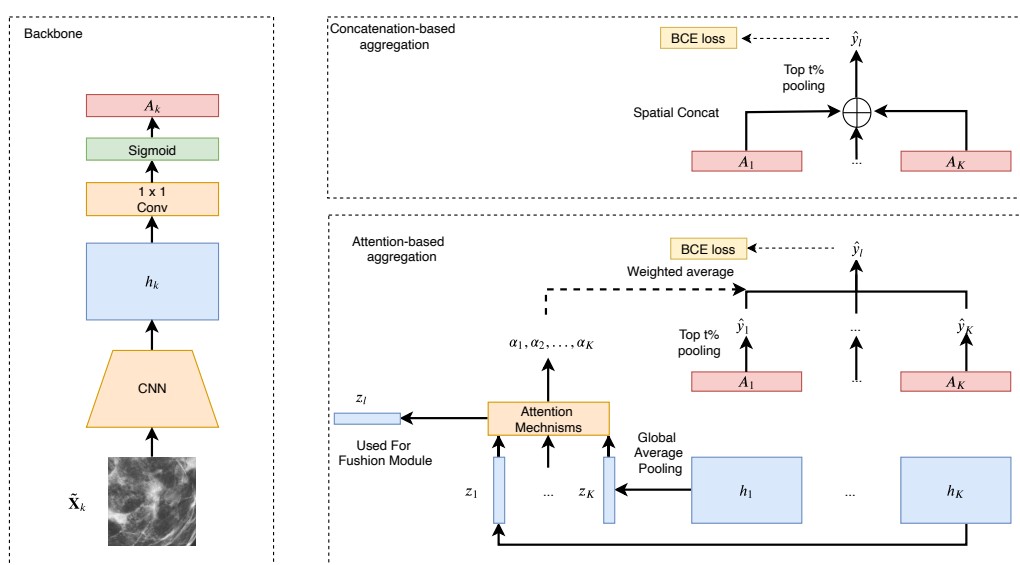

Figure 7: Architecture of the local module. The backbone network (left) is applied to each of the selected patches $\tilde{\mathbf{x}}_k$ ($k \in 1, \ldots, K$) to extract a patch-level saliency map $\mathbf{A}_k$ and a feature map $h_k$. The patch-level saliency maps can be combined using two aggregation strategies: (1) *Concatenation-based aggregation* (top right), where we concatenate the saliency maps spatially and apply $\mathrm{top}\, t\%$ pooling. (2) *Attention-based aggregation* (bottom right), where $\mathrm{top}\, t\%$ pooling is used to obtain patch-level predictions $\hat{y}_k$ from each patch, which are then combined using attention weights $\alpha_i \in [0, 1]$ computed by the gated attention mechanism (Ilse et al., 2018). That is, the classification prediction is computed as $\hat{y}_l = \sum_{i=1}^{K} \alpha_i \hat{y}_i$, and the representation vector as $z_l = \sum_{i=1}^{K} \alpha_i z_i$.

of $t$ in the $\mathrm{top}\, t\%$ pooling step in Appendix G.7. The selected value for both aggregation strategies is 20.

## Appendix D. Localization and classification

A few works (Choe et al., 2020; Choe and Shim, 2019; Singh and Lee, 2017; Yao et al., 2018) observed that localization and classification accuracy of weakly-supervised models are not perfectly correlated. An empirical study by Choe et al. (2020) suggested that the best localization performance is usually achieved at early epochs of training, when the classifier is not fully trained. Furthermore, it has been observed that classification performance does not necessarily correlate with the localization performance across different architectures (Choe et al., 2020). We made similar observations in our work. Our local module reduces the stride of the convolution kernel to preserve more spatial resolution, which results in a smaller receptive field and has some adversarial effects for classification performance. Therefore, only the global module classification prediction is used as the classification result for our system. We provide the segmentation and classification results of different local module architectures in Appendix G.3.

## Appendix E.  Evaluation metrics

**Dice**   The Dice similarity coefficient is computed as $\text{Dice} = (2 \times \mathbf{S} \times \mathbf{G})/(\mathbf{S}^2 + \mathbf{G}^2)$, where $\mathbf{S}$ is the saliency map produced by the model, and $\mathbf{G}$ is the ground truth binary mask.

**PxAP**   We measure the pixel-wise precision and recall using the pixel average precision (PxAP) metric introduced by Choe et al. (2020). PxAP is the area under the pixel precision-recall curve, as we vary a threshold $\tau$ that determines precision-recall trade-off. We define the pixel precision and recall at threshold $\tau$ as:

$$\text{PxPrec}(\tau) = \frac{|\{\mathbf{S}^c(i,j) \geq \tau\} \cap \{\mathbf{G}^c(i,j) = 1\}|}{|\{\mathbf{S}^c(i,j) \geq \tau\}|}$$
$$\text{PxRec}(\tau) = \frac{|\{\mathbf{S}^c(i,j) \geq \tau\} \cap \{\mathbf{G}^c(i,j) = 1\}|}{|\{\mathbf{G}^c(i,j) = 1\}|},$$

where $\mathbf{S^c}$ is the saliency map produced by the model, $\mathbf{G^c}$ is the ground truth binary mask, $i, j$ are the spatial indices, and $c$ is the class. PxAP is defined as:

$$\text{PxAP} := \sum_l \text{PxPrec}\left(\tau_l\right)\left(\text{Px Rec}\left(\tau_l\right) - \text{PxRec}\left(\tau_{l-1}\right)\right).$$

PxAP can be computed in two ways:

1. *Image-level PxAP:* We compute PxAP for each image separately, then we compute the mean and standard deviation over all images. The advantage of this metric is that we put the same emphasis on the images with large lesions and the images with small lesions.

2. *Dataset-level PxAP:* We aggregate the pixels from all images to obtain a single dataset pixel precision-recall curve and compute a single PxAP. Since the threshold to compute the precision and recall is applied to the whole dataset, the advantage is that it can help users to choose the preferred operating threshold $\tau$ that provides the best precision-recall trade-off for their downstream applications.

## Appendix F.  Hyperparameter tuning

To compare fairly between model architectures, we follow a similar hyperparameter tuning procedure as Shen et al. (2021). We optimize the hyperparameters with random search (Bergstra and Bengio, 2012) for both baselines and for GLAM. We search for the learning rate $\eta \in 10^{[-5.5,-4]}$ on a logarithmic scale. We also search for the regularization weight $\lambda \in 10^{[-5.5,-3.5]}$ on a logarithmic scale. We search for the pooling threshold $t \in \{1\%, 2\%, 3\%, 5\%, 10\%, 20\%\}$ for GMIC (Shen et al., 2021) and for GLAM. The pooling function in the original CAM (Zhou et al., 2016) is global average pooling, which is an extreme version of $\text{top } t\%$, where $t = 100\%$. In order to provide a fair comparision, we also tune the pooling threshold $t \in \{1\%, 2\%, 3\%, 5\%, 10\%, 20\%, 100\%\}$ for this model.

In all cases, we train 30 separate models using hyperparameters randomly sampled from the ranges described above. We train the models (the competing methods and our global module) for 50 epochs and select the weights from the training epoch that achieves the highest validation segmentation performance based on Dice score. We then fix the global module and conduct the

local module training based on the patch proposals selected according to the global module. The local module converges faster, so we train the local model for 20 epochs and select the weights from the training epoch that achieves the highest validation segmentation performance according to Dice score. Based on the trained global and local module, the joint training takes less than 5 epochs. All experiments are conducted on an NVIDIA Tesla V100 GPUs. The global module takes about 2 days to train and the local module takes about 4 days.

## Appendix G. Ablation studies

### G.1. Importance of using multiple-scale feature maps in the global module

In the following table, we compare the performance of the individual saliency maps of the global module with the combined saliency map $\mathbf{S}_g$ (see Appendix B). The latter has much better segmentation performance.

| Saliency map | Dice | | image-level PxAP | | dataset-level PxAP | |
|---|---|---|---|---|---|---|
| | Malignant | Benign | Malignant | Benign | Malignant | Benign |
| $\mathbf{S}_g$ | **0.325 ± 0.239** | **0.261 ± 0.185** | **0.363 ± 0.276** | **0.302 ± 0.266** | **0.324** | **0.140** |
| $\mathbf{S}_0$ | 0.155 ± 0.192 | 0.130 ± 0.108 | 0.139 ± 0.184 | 0.121 ± 0.175 | 0.117 | 0.049 |
| $\mathbf{S}_1$ | 0.319 ± 0.238 | 0.240 ± 0.190 | 0.318 ± 0.257 | 0.269 ± 0.258 | 0.254 | 0.113 |
| $\mathbf{S}_2$ | 0.213 ± 0.209 | 0.157 ± 0.150 | 0.220 ± 0.211 | 0.113 ± 0.149 | 0.224 | 0.060 |

### G.2. Selection of training data for the local module

In order to generate the training data for the local module, we use patches containing lesions (*positive examples*), which are obtained from the global module, and patches that do not contain lesions (*negative examples*), which are randomly sampled from images that do not contain lesions. Alternatively, one could generate negative examples by using the output patches generated by the global module from scans without lesions. The following table shows that this results in significantly worse performance.

| Random sampling negative examples | Dice | | image-level PxAP | | dataset-level PxAP | |
|---|---|---|---|---|---|---|
| | Malignant | Benign | Malignant | Benign | Malignant | Benign |
| No | 0.184 ± 0.192 | 0.188 ± 0.205 | 0.282 ± 0.230 | 0.187 ± 0.202 | 0.126 | 0.064 |
| Yes | **0.343 ± 0.283** | **0.297 ± 0.287** | **0.444 ± 0.337** | **0.337 ± 0.310** | **0.207** | **0.126** |

### G.3. Architecture of the local module

In this ablation study, we investigate the influence of the design of the local module on its segmentation and classification performance. We compare Resnet-34, a ResNet-18 with reduced stride in the residual blocks (ResNet-18-HR) and the proposed local module, which consists of a Resnet-34 with reduced stride in the residual blocks (Resnet-34-HR). Recall that the motivation to reduce the stride is to produce a saliency map with higher resolution, as explained in Appendix C. The results are shown in the following table. We see that carefully selecting the backbone architecture is crucial to ensure a good segmentation performance of the local module. It is worth mentioning that reducing the stride decreases the classification performance of the local module (which is why we do not use its output to perform classification).

| Network Architecture | Dice Malignant | Benign | image-level PxAP Malignant | Benign | dataset-level PxAP Malignant | Benign | classification AUC Malignant | Benign |
|---|---|---|---|---|---|---|---|---|
| ResNet-34 | $0.163 \pm 0.224$ | $0.218 \pm 0.249$ | $0.272 \pm 0.257$ | $0.256 \pm 0.268$ | 0.097 | 0.067 | 0.826 | 0.705 |
| ResNet-18-HR | $0.226 \pm 0.243$ | $0.172 \pm 0.235$ | $0.330 \pm 0.301$ | $0.208 \pm 0.260$ | 0.137 | 0.077 | 0.706 | 0.657 |
| ResNet-34-HR | $\mathbf{0.343 \pm 0.283}$ | $\mathbf{0.297 \pm 0.287}$ | $\mathbf{0.444 \pm 0.337}$ | $\mathbf{0.337 \pm 0.310}$ | **0.207** | **0.126** | 0.747 | 0.682 |

## G.4. Number of patches used in the local module during training

To study the impact of the number of patches used in the local module for training. We freeze the global module and train three models where the local module receives one, three or six patches respectively.[1] The performance of these models is shown in the following table. The local module achieves better segmentation performance when more patches are used during training, which is consistent with results reported by Shen et al. (2021).

| patch number | Dice Malignant | Benign | image-level PxAP Malignant | Benign | dataset-level PxAP Malignant | Benign |
|---|---|---|---|---|---|---|
| 1 | $0.268 \pm 0.229$ | $0.280 \pm 0.267$ | $0.365 \pm 0.280$ | $0.237 \pm 0.239$ | 0.153 | 0.107 |
| 3 | $0.320 \pm 0.269$ | $0.285 \pm 0.282$ | $0.420 \pm 0.318$ | $0.283 \pm 0.276$ | 0.197 | 0.111 |
| 6 | $\mathbf{0.343 \pm 0.283}$ | $\mathbf{0.297 \pm 0.287}$ | $\mathbf{0.444 \pm 0.337}$ | $\mathbf{0.337 \pm 0.310}$ | **0.207** | **0.126** |

## G.5. Number of patches used in the local module during inference

In this ablation study we use the model trained with six patches and evaluate its performance when using different numbers of patches during inference. The scatter plot in Figure 8 compares the segmentation performance when using one patch and when using three patches for 400 example images from the validation set. Using one patch leads to better performance for most of the images, but completely fails for some. Failure may occur when the patch selected by the global module for the positive example does not contain lesions. For our dataset, using one patch produces better segmentation performance overall, but this may not be the case for other applications (e.g. if there are a large number of lesions in each image).

## G.6. Fusion module

In this ablation study, we compare the proposed model to two different models without fusion modules. In Model 1 the local module is trained using concatenation-based aggregation, in Model 2 it is trained with attention-based aggregation (see Section 3.2). To ensure a fair comparison, the models are obtained by jointly training the same pretrained global and local modules. We report the results in the following table. $\mathbf{S}_{full}$ denotes the combined saliency map from the proposed model. $\mathbf{S}_{a1}$ denotes the saliency map from Model 1. $\mathbf{S}_{a2}$ denotes the saliency map from Model 2. Incorporating a fusion module results in better localization performance.

| | Dice Malignant | Benign | image-level PxAP Malignant | Benign | dataset-level PxAP Malignant | Benign |
|---|---|---|---|---|---|---|
| $\mathbf{S}_{full}$ | $\mathbf{0.390 \pm 0.253}$ | $\mathbf{0.335 \pm 0.259}$ | $\mathbf{0.461 \pm 0.296}$ | $\mathbf{0.396 \pm 0.315}$ | **0.341** | **0.215** |
| $\mathbf{S}_{a1}$ | $0.318 \pm 0.252$ | $0.324 \pm 0.263$ | $0.447 \pm 0.298$ | $0.373 \pm 0.306$ | 0.308 | 0.197 |
| $\mathbf{S}_{a2}$ | $0.376 \pm 0.259$ | $\mathbf{0.335 \pm 0.260}$ | $0.459 \pm 0.297$ | $0.393 \pm 0.313$ | 0.331 | 0.210 |

1. Due to GPU memory constraints, the maximum number of patches we are able to train with is six.

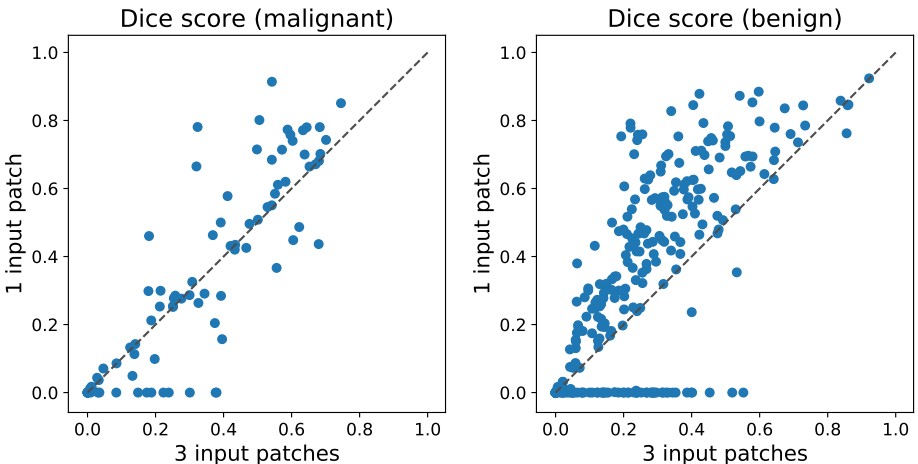

Figure 8: Dice score of the local module when using one or three patches as its input during infer-
ence. The results correspond to 400 examples from the validation set. Using one patch
leads to better performance for most of the images, but completely fails for a some. Fail-
ure may occur when the patch selected by the global module for the positive example
does not contain lesions.

### G.7. Impact of top $t\%$ pooling

As already shown by Shen et al. (2021), the choice of pooling hyperparameter has a significant
impact on localization performance. In order to study how this hyperparameter influences the
localization ability of the local module, we experimented with five choices of *top $t\%$ pooling*:
$t \in \{10, 15, 20, 30, 60\}$. For all of them, we use the same trained global module, and train the local
module without joint training. In the following table, we report the segmentation score achieved by
each value of $t$ on the test set. The results indicate that the pooling hyperparameter has a significant
impact on the localization performance of the patch-level network (*i.e.* the local module), and the
optimal $t$ may be different for different classes.

| | Dice | | image-level PxAP | | dataset-level PxAP | |
|---|---|---|---|---|---|---|
| *top $t\%$* | Malignant | Benign | Malignant | Benign | Malignant | Benign |
| 0.1 | $0.280 \pm 0.268$ | $0.284 \pm 0.279$ | $0.419 \pm 0.325$ | $0.319 \pm 0.297$ | 0.169 | 0.106 |
| 0.15 | $0.314 \pm 0.269$ | $0.290 \pm 0.283$ | $0.408 \pm 0.315$ | $0.300 \pm 0.284$ | 0.158 | 0.113 |
| 0.2 | $0.343 \pm 0.283$ | $\mathbf{0.297 \pm 0.287}$ | $\mathbf{0.444 \pm 0.337}$ | $\mathbf{0.337 \pm 0.310}$ | 0.207 | **0.126** |
| 0.3 | $\mathbf{0.353 \pm 0.277}$ | $0.281 \pm 0.280$ | $0.425 \pm 0.319$ | $0.310 \pm 0.297$ | **0.212** | 0.113 |
| 0.6 | $0.307 \pm 0.259$ | $0.260 \pm 0.269$ | $0.408 \pm 0.314$ | $0.334 \pm 0.316$ | 0.221 | 0.114 |

### G.8. Impact of the $\lambda$ hyperparameter associated with $\ell_1$ regularization

To investigate the influence of the $\lambda$ hyperparameter associated with $\ell_1$ regularization, we con-
duct the following experiments for the global module. We fixed all other hyper-parameter to
the same values as used in the optimal model. To keep the comparison scenario simple, the

saliency map of the global module is computed as $\mathbf{S}_g = \frac{\mathbf{S}_0 + \mathbf{S}_1 + \mathbf{S}_2}{3}$. $\lambda$ is randomly selected from $10^{[-5.5, -3.5]}$ on a logarithmic scale during hyper-parameter searching for training, we take the following samples of $\lambda$ for ablation study by randomly selection within the searching space: $10^{-3.00}, 10^{-3.53}, 10^{-3.81}, 10^{-4.24}, 10^{-4.73}$ and report their Dice score on the test set. From the following table, we see that the performance of the saliency map is stable within the range of our hyper-parameter search space.

| | Dice | |
| --- | --- | --- |
| $\log \lambda$ | Malignant | Benign |
| -3.00 | $0.294 \pm 0.202$ | $0.197 \pm 0.124$ |
| -3.53 | $0.255 \pm 0.209$ | $0.211 \pm 0.143$ |
| -3.81 | $0.286 \pm 0.207$ | $0.215 \pm 0.141$ |
| -4.24 | $0.279 \pm 0.225$ | $0.240 \pm 0.160$ |
| -4.73 | $\mathbf{0.304 \pm 0.211}$ | $\mathbf{0.246 \pm 0.163}$ |

### G.9. An alternative approach to aggregating local and global saliency maps

For simplicity, in GLAM we set $\mathbf{S}_c = (\mathbf{S}_g + \mathbf{S}_l)/2$. Here, we investigate applying a weighted average $\mathbf{S}_c = \gamma_c \mathbf{S}_g + (1 - \gamma_c)\mathbf{S}_l$, with an additional hyperparameter $\gamma_c \in [0, 1]$. The results in the following table show that this may slightly improve performance for some metrics.

| | Dice | | image-level PxAP | | dataset-level PxAP | |
| --- | --- | --- | --- | --- | --- | --- |
| $\gamma_c$ | Malignant | Benign | Malignant | Benign | Malignant | Benign |
| 0 | $0.406 \pm 0.266$ | $0.297 \pm 0.270$ | $0.451 \pm 0.306$ | $0.318 \pm 0.282$ | 0.269 | 0.158 |
| 0.1 | $0.410 \pm 0.266$ | $0.310 \pm 0.272$ | $\mathbf{0.470 \pm 0.300}$ | $0.390 \pm 0.310$ | 0.321 | 0.202 |
| 0.2 | $\mathbf{0.411 \pm 0.265}$ | $0.322 \pm 0.272$ | $0.469 \pm 0.300$ | $0.391 \pm 0.210$ | 0.331 | 0.208 |
| 0.3 | $0.408 \pm 0.263$ | $0.330 \pm 0.270$ | $0.467 \pm 0.299$ | $0.393 \pm 0.311$ | 0.336 | 0.212 |
| 0.4 | $0.401 \pm 0.258$ | $\mathbf{0.335 \pm 0.265}$ | $0.465 \pm 0.298$ | $0.393 \pm 0.312$ | 0.340 | 0.214 |
| 0.5 | $0.390 \pm 0.253$ | $\mathbf{0.335 \pm 0.259}$ | $0.461 \pm 0.296$ | $\mathbf{0.396 \pm 0.315}$ | $\mathbf{0.341}$ | $\mathbf{0.215}$ |
| 0.6 | $0.373 \pm 0.246$ | $0.332 \pm 0.251$ | $0.447 \pm 0.291$ | $0.390 \pm 0.312$ | 0.336 | 0.207 |
| 0.7 | $0.351 \pm 0.239$ | $0.320 \pm 0.242$ | $0.434 \pm 0.286$ | $0.380 \pm 0.307$ | 0.329 | 0.198 |

