# OpenReview forum: "Weakly-supervised High-resolution Segmentation of Mammography Images for Breast Cancer Diagnosis"
_MIDL.io/2021/Conference — MIDL 2021_

### Official Review · AnonReviewer3 · 2021-02-26

**Confidence:** 4
**Preliminary Rating:** 3
**Recommendation:** Poster
**Final Rating:** 3

**Summary:**

This paper proposes a weakly-supervised method for image segmentation using only image-level labels. The main idea is to select informative regions ( patches ) that may contain ROIs via coarse-level localization and then to perform segmentation on selected patches rather than the entire image. The proposed GLAM produces fine-grained saliency maps with a resolution that is 300 times higher than that of previous works ( Shen et al., 2021 ). The authors show that the saliency map generated by GLAM enables high-resolution segmentation of lesions relevant to the breast cancer diagnosis.

**Strengths:**

1. The proposed model can process high-resolution medical images in a memory-efficient way.
2. The paper is well-written and easy to follow.
3. The method is evaluated on a dataset containing more than one million mammography images.
4. The model outperforms existing baselines in the segmentation of both benign and malignant lesions, improving the similarity score relatively by 39.6 % and 20 %, respectively. At the same time, it preserves classification accuracy.

**Weaknesses:**

Overall, this paper is good enough, and I only have the following concern:
The proposed method does not compare with enough weakly supervised segmentation methods, and it would be better to make these comparisons in the final version.

**Deanonymize Review:**

no

**Final Rating Justification:**

The response addresses my concerns and I stick to my rating decision.

**Justification Of The Preliminary Rating:**

This paper presents a method for weakly-supervised segmentation of high-resolution medical images. The authors apply this model to the breast cancer diagnosis with screening mammography and validate it on a large clinically-realistic dataset. The paper is well-written and easy to follow and the proposed method is efficient and effective.

**Paper Type:**

validation/application paper

**Questions To Address In The Rebuttal:**

Please see "Weaknesses".

**Special Issue:**

no

---

> ### Author Response · Authors · 2021-03-17
> **Response to AnonReviewer3**
>
> We thank the reviewer for the valuable comments.
>
> > Overall, this paper is good enough, and I only have the following concern: The proposed method does not compare with enough weakly supervised segmentation methods, and it would be better to make these comparisons in the final version.
>
> This is an excellent point. The reason why we decided to use the two baselines in the paper is that GMIC[1] is optimized for high-resolution medical images, and CAM[2] has been recently shown to outperform most existing methods on segmentation of natural images in a recent comprehensive empirical study by Choe et al. [3] when evaluated on completely held-out data. We will make this clear in the revised version of the paper. However, if the reviewer has any other specific suggestions, we would be happy to incorporate comparisons to them.
>
> [1] Shen, Y., Wu, N., Phang, J., Park, J., Liu, K., Tyagi, S., Heacock, L., Kim, S.G., Moy, L., Cho, K. and Geras, K.J.. An interpretable classifier for high-resolution breast cancer screening images utilizing weakly supervised localization. Medical image analysis, 2021.
>
> [2] Zhou, A. Khosla, A. Lapedriza, A. Oliva, and A. Torralba. Learning deep features for discriminative localization. In CVPR, 2016.
>
> [3] Junsuk Choe, Seong Joon Oh, Seungho Lee, Sanghyuk Chun, Zeynep Akata, and Hyunjung Shim. Evaluating weakly supervised object localization methods right. In CVPR, 2020.

---

### Official Review · AnonReviewer2 · 2021-03-05

**Confidence:** 4
**Preliminary Rating:** 3
**Recommendation:** Poster
**Final Rating:** 3

**Summary:**

This paper presents a new weakly-supervised learning framework to detect lesion regions from high-resolution mammography images. They first use a global module to select ROIs via coarse-level localization and then use a local module to conduct fine-grained segmentation from the image patches. The proposed method is evaluated on a large clinically-realistic dataset and achieves much better performance than other methods.

**Strengths:**

1. The problem of detecting lesions from high-resolution images with the image-level label is important yet difficult. The proposed two-stage pipeline is reasonable and achieves better performance on a large-scale dataset.
2. The authors conduct extensive experiments and ablation studies to show the effectiveness of each design components.

**Weaknesses:**

1. The authors put some important technical details into supplementary materials and cause some difficulty to fully understand the technical details of the whole framework.

2. Since the whole framework has two steps and the second step (i.e., local module training) relies on the first step model. There are maybe some difficulties to effectively train the whole framework. For example, the labels of the extracted patches for local module training may be noise. In this case, how do you ensure the robustness of the local model?

**Deanonymize Review:**

no

**Detailed Comments:**

Following are some detailed comments.
1. Why do you use top t% pooling to get the image-level classification prediction from the saliency map? What is the advantage of it?

2. When calculating the dice metric, we need to use the binary segmentation mask.  How do you get the binary mask from the saliency map? Do you use a thresholding-based method?

3. Have you checked the extracted patches for local module training? Is there any mistakes for the positive examples?


**Final Rating Justification:**

The authors have solved most of my questions. I tend to accept this manuscript.


**Justification Of The Preliminary Rating:**

In general, this is a good paper and presents an effective solution to detect lesion regions from high-resolution mammography images. The coarse-to-fine detection procedure is reasonable. The authors also conduct extensive experiments to show the effectiveness of the proposed method.

**Paper Type:**

both

**Questions To Address In The Rebuttal:**

See Weakness and Detailed Comments.

**Special Issue:**

no

---

> ### Author Response · Authors · 2021-03-17
> **Response to AnonReviewer2**
>
> We thank the reviewer for the helpful comments.
>
> > The authors put some important technical details into supplementary materials and cause some difficulty to fully understand the technical details of the whole framework.
>
> We thank the reviewer for pointing this out. We will rearrange the main paper and the Appendix to provide a more self-contained technical description in the final version of the paper.
>
> > Since the whole framework has two steps and the second step (i.e., local module training) relies on the first step model. There are maybe some difficulties to effectively train the whole framework. For example, the labels of the extracted patches for local module training may be noise. In this case, how do you ensure the robustness of the local model?
>
> This is a good point. The input of the local module does indeed depend on the saliency map of the global module. For this reason, we train the model sequentially, to ensure that the global module is producing reasonable outputs while training the local module, as explained in Section 3.3. In addition, the patch aggregation function in the local module aggregates multi-patch information to give an image-level prediction, taking into account that some of the patches selected by the global module may not contain lesions even if the image-level label is positive. This further bolsters the robustness of the local module. We will make sure to clarify this in the final version of the paper.
>
> > Why do you use top t% pooling to get the image-level classification prediction from the saliency map? What is the advantage of it?
>
> This is a good question, which we will discuss in more detail in the revised version. In the weakly supervised localization paradigm, the pooling stage is necessary because we only have access to image-level labels. Several pooling strategies have been proposed in the literature, including global average pooling [1], global maximum pooling [2], log-sum-exp pooling [3,4]. Top t% pooling has been shown to be effective in previous work [5], and it has the advantage of being flexible. We would be able to adjust the value of the hyperparameter t, which is set based on validation performance. It reduces to global average pooling when t is set to 100% and to global maximum pooling when t is very small (1 divided by the number of pixels in the saliency map). In Appendix H.7. we provide an ablation study on the performance of different t% values.
>
> > When calculating the dice metric, we need to use the binary segmentation mask. How do you get the binary mask from the saliency map? Do you use a thresholding-based method?
>
> Following previous work [5], our Dice metric is evaluated on the probabilistic saliency map rather than the binarized one. Specifically, it is computed as Dice = (2 * S * G)/(S^2 + G^2), where  S is the probabilistic saliency map produced by the model, and   G is the ground truth binary mask. It naturally takes into account the predicted values while avoiding selecting the discretization threshold. The values we report are computed as mean and standard deviation~(std) over images for which segmentation labels are available. The PxAP, on the other hand, aims to compute the area under the pixel precision-recall curve as we vary a threshold that determines precision-recall trade-off. Details on the different metrics are in Appendix E.
>
> > Have you checked the extracted patches for local module training? Is there any mistakes for the positive examples?
>
> This is a great suggestion. We have done some additional analysis to investigate this. We used a set of images with tumors from the validation set, and checked the 3 first patches extracted by the global module. These patches did not contain tumors only in 40 out of 300 benign images (13%) and 6 out of 75 malignant images (8%). We will add this analysis to the paper.
>
> [1] Zhou, A. Khosla, A. Lapedriza, A. Oliva, and A. Torralba. Learning deep features for discriminative localization. In CVPR, 2016.
>
> [2] M. Oquab, L. Bottou, I. Laptev, and J. Sivic.  Is object localization for free? weakly-supervised learning with convolutional neural networks. In CVPR, 2015.
>
> [3] P. O. Pinheiro and R. Collobert.  From image-level to pixel-level labeling with convolutional networks.  In CVPR, 2015.
>
> [4] Durand, T., Mordan, T., Thome, N., Cord, M.. Wildcat: Weakly supervised learning of deep convnets for image classification, pointwise localization and segmentation, In CVPR 2017.
>
> [5] Shen, Y., Wu, N., Phang, J., Park, J., Liu, K., Tyagi, S., Heacock, L., Kim, S.G., Moy, L., Cho, K. and Geras, K.J.. An interpretable classifier for high-resolution breast cancer screening images utilizing weakly supervised localization. Medical image analysis, 2021.

---

### Official Review · AnonReviewer4 · 2021-03-09

**Confidence:** 4
**Preliminary Rating:** 2
**Final Rating:** 2

**Summary:**

The paper presents a weakly-supervised method for segmenting and classifying high-resolution mammography images. The proposed architecture, called GLAM, is composed of three modules: a global module using a low capacity network to obtain a coarse segmentation,  a local module which takes high-resolution patches having largest global saliency and outputs fine grained saliency maps, and a fusion module to predict the final diagnosis label using the latent representation of the global and local modules. The proposed GLAM method is tested on the NYU Breast Cancer Screening Dataset and compared against GMIC, CAM and a U-Net trained in a supervised manner. Results show GLAM to outperform GMIC and CAM in terms of segmentation and classification accuracy.

**Strengths:**

* The paper is well written and easy to follow

* The method is evaluated on a large dataset and shows statistically significant improvements in term of segmentation accuracy compared to CAM and GMIC.

* Experiment are very comprehensive and include several ablation studies evaluating the impact of using multiple-scale feature maps in the global module, randomly sampling patches from negative examples, choice of network, number of patches in the local module, etc.


**Weaknesses:**

* The main weakness it the lack of clear methodological contributions compared to previous works. In particular, the proposed method is very similar to (Shen et al, 2021) which also includes global, local and fusion modules, and aims to classify high-resolution mammography images.

* Results are mixed. While GLAM outperforms CAM and GMIC in terms of segmentation, classification accuracy is worse. For segmentation, it is unclear whether the performance of the method (Dice of 0.390 and 0.335) is sufficient for real-life clinical applications.


**Deanonymize Review:**

no

**Detailed Comments:**

* The definition of weakly supervised learning (WSL) for segmentation in the introduction is somewhat limited. WSL normally includes other types of noisy or incomplete annotations like scribbles, bounding boxes, etc.

* The difference between the proposed method and (Shen et al, 2021) is not clear to me. Contrary to what is mentioned in the paper, this previous works also works with high-resolution mammography images (2944×1920) and uses a local module with high-capacity to extract fine-grained visual details from extracted regions.

* I may have missed this information in the ablation studies, but what is the purpose and impact of the l1 sparsifying loss ?

**Final Rating Justification:**

The paper has some merits, however I still fell that the contributions w.r.t. previous work (GLAM) are somewhat incremental and that the low segmentation accuracy limits its practical use.

**Justification Of The Preliminary Rating:**

The methodological novelty of the proposed method is unclear. In particular, the overall architecture and evaluation setup is very similar to (Shen et al, 2021). The practical advantages of this method also need to be clarified.

**Paper Type:**

methodological development

**Questions To Address In The Rebuttal:**

* Clarify the novel contributions compared to  (Shen et al, 2021)

* Better motivation the practical value of the method, given that the Dice score are low and classification accuracy is no better than baselines.

**Special Issue:**

no

---

> ### Author Response · Authors · 2021-03-17
> **Response to AnonReviewer4(Part One)**
>
> We thank the reviewer for the constructive comments.
>
> > The main weakness it the lack of clear methodological contributions compared to previous works. In particular, the proposed method is very similar to (Shen et al, 2021) which also includes global, local and fusion modules, and aims to classify high-resolution mammography images.
>
> This is an important point. We thank the reviewer for raising it, and we will include a detailed explanation of the differences between GMIC [1] and GLAM in the final version.
>
> The different modules of GMIC are indeed similar to those of GLAM, but crucially GMIC is not capable of generating high-resolution saliency maps. The critical difference is that the local module in GLAM is explicitly designed to produce these high-resolution saliency maps, whereas in GMIC the local module does not include a saliency map (and even if one just used the deeper feature maps to try to obtain a visualization, they would still be low resolution.  Even if the resolution requirement is fullfilled, there are also other important elements needed for satisfactory segmentation performance as described below). Designing a local module capable of performing segmentation requires training it using an aggregation function to exploit image-level labels (because patch-level labels are not available). To the best of our knowledge, training saliency maps hierarchically in this way is a novel contribution (that is definitely not present in GMIC). In addition,optimizing the parameters of the network in this novel framework is not trivial. There are several elements that are essential to achieve good segmentation performance, including the selection of negative patches to train the local module (which should be done randomly from images without lesions as described in Appendix H.2), the architecture choices in the local module (see Appendix H.3), the patch-aggregation strategy used to train the local module (see Appendix H.7) and the choice of the number of patches used during training and inference in the local module (see Appendices H.4 and H.5). The influence of these elements in segmentation performance are not considered in GMIC, or in any previous works (to the best of our knowledge).
>
> It is also worth noting that the global module in GLAM is also different from the global module in GMIC: it has a multi-scale structure, as explained in Appendix B, which improves performance as reported in Appendix H.1.
>
>
> > Results are mixed. While GLAM outperforms CAM and GMIC in terms of segmentation, classification accuracy is worse.
>
> This is another important point. Our focus in this work is to optimize segmentation performance while maintaining a similar classification accuracy (GLAM has an AUC of 0.770 for benign lesions, compared to 0.770 for CAM and 0.780 for GMIC, and an AUC of 0.882 for malignant lesions, compared to 0.894 for CAM and 0.886 for GMIC). It is important to note that these models have been optimized according to their segmentation performance. As discussed in Appendix D and in previous literature (e.g. [1,2,3,4,5]), within the weakly supervised localization paradigm, classification and segmentation accuracy are often not necessarily correlated. We report the segmentation and classification performance corresponding to different local module architectures in Appendix H.3 to illustrate this. We will further highlight this point in the final version of the paper.
>
> > For segmentation, it is unclear whether the performance of the method (Dice of 0.390 and 0.335) is sufficient for real-life clinical applications.
>
> The question of what segmentation performance is sufficient for real-world applications is very interesting. The answer would depend on the specific situation. Automatic segmentation can be used as an aid for radiologists, enabling them to analyze mammography exams more efficiently and accurately. It also has the potential to reveal occult lesions that could be overlooked by humans. Whether such an improvement in segmentation accuracy will indeed translate into an improvement in clinical practice, is a question that can only be answered by a future clinical study. We will add a discussion about this in the final version of the paper. We would also like to point out that besides clinical utility for this particular application, we view our model as a methodological contribution.
>
> > The definition of weakly supervised learning (WSL) for segmentation in the introduction is somewhat limited. WSL normally includes other types of noisy or incomplete annotations like scribbles, bounding boxes, etc.
>
> We agree with the reviewer and will edit the definition accordingly.

---

> ### Author Response · Authors · 2021-03-17
> **Response to AnonReviewer4(Part Two)**
>
> > The difference between the proposed method and (Shen et al, 2021) is not clear to me. Contrary to what is mentioned in the paper, this previous works also works with high-resolution mammography images (2944×1920) and uses a local module with high-capacity to extract fine-grained visual details from extracted regions.
>
> We thank the reviewer for highlighting that the differences are not sufficiently emphasized. We have explained the differences in detail in Response to AnonReviewer4(Part One) and will make this very clear in the final version.
>
> > I may have missed this information in the ablation studies, but what is the purpose and impact of the l1 sparsifying loss ?
>
> The purpose of this element of the training loss is to promote sparsity in the saliency maps. This is necessary because lesion sizes tend to be small, as shown in previous works (e.g. [1]). We will add an ablation study to evaluate the robustness of the results with respect to the sparsity parameter $\lambda$ in the final version of the paper.
>
> [1] Shen, Y., Wu, N., Phang, J., Park, J., Liu, K., Tyagi, S., Heacock, L., Kim, S.G., Moy, L., Cho, K. and Geras, K.J.. An interpretable classifier for high-resolution breast cancer screening images utilizing weakly supervised localization. Medical image analysis, 2021.
>
> [2] Junsuk Choe, Seong Joon Oh, Seungho Lee, Sanghyuk Chun, Zeynep Akata, and Hyunjung Shim. Evaluating weakly supervised object localization methods right. CVPR 2020.
>
> [3] Junsuk Choe and Hyunjung Shim. Attention-based dropout layer for weakly supervised object localization. CVPR 2019.
>
> [4] Krishna Kumar Singh and Yong Jae Lee. Hide-and-seek: Forcing a network to be meticulous for weakly-supervised object and action localization. ICCV 2017
>
> [5] Li Yao, Jordan Prosky, Eric Poblenz, Ben Covington, and Kevin Lyman. Weakly supervised medical diagnosis and localization from multiple resolutions. preprint arXiv:1803.07703

---

### Official Review · AnonReviewer1 · 2021-03-09

**Confidence:** 4
**Preliminary Rating:** 3
**Recommendation:** Oral
**Final Rating:** 4

**Summary:**

This paper tackles the important topic of tumor segmentation in breast scans, using only tags as labels during training. This setting (which is some form of multiple instance learning) is one of the "weakest" setting in weakly supervised segmentation, and as such the trickiest.

The authors propose an elegant two-stage method, based on saliency maps, that is able to produce a segmentation map with a much higher resolution compared to existing methods.

The authors manages to get approximately 80% of the performances of a fully-supervised U-Net, which is very impressive given how little annotated data they use.

**Strengths:**

- Higher resolution than existing method, which is very important in the clinical setting as some tumors can be very small
- The authors compare to two other weakly supervised baseline, and to one fully supervised upper bound
- 4 different reported metrics
- The discussion on the strength and weaknesses of each modules (section _segmentation properties of the global and local saliency maps_) is interesting and explains well the design choice to combine the two outputs.
- Additional ablation studies in the appendix

**Weaknesses:**

- I think the paper would benefit from mentioning that the current task is a case of multiple instance learning [1]. Not because its solutions are relevant here (it's not), but because it gives I think an interesting and relevant framing to tackle the task. This could be especially relevant in a future extension when trying to increase the resolution further. (For instance, by treating the patches selected after the first stage as positive or negative "bags", it could be plugged into a regular network utilizing such information; for instance by using a prior on how much the positive bags are filled with tumor.)
- I think a discussion on the training and inference cost is missing, especially as two full networks are involved. A higher cost will absolutely not invalidate the authors method, but it would give some perspective.
- The section on the aggregation function, to train the local module, should be clearer and more prominent in the paper, as it is, I believe, the main ingredient to the method's success. Currently it is scattered across the paper when it should be one dedicated subsection in the main-matter.



[1] Maron, O., & Lozano-Pérez, T. (1998). A framework for multiple-instance learning. Advances in neural information processing systems, 570-576.

**Deanonymize Review:**

no

**Detailed Comments:**

Minor and in no particular order:

- The paper is a big long, compared to the MIDL limit of 8 pages, and a lot of the main content ends up in the appendix. But, as I personally disagree with this year's 8 pages limit, this isn't an issue to me. Nevertheless, the camera ready version could probably benefit from some reorganization between main-matter and appendix.
- Your tables (such as table 1) could probably be made a tiny bit more readable, by using the features of the package `booktabs`. The table columns could be swapped from `llllllllll` to `lcccccccc` and the second `\hline` replaced by `\cmidrule(lr){2-3}\cmidrule(lr){4-5}\cmidrule(lr){6-7}\cmidrule(lr){8-9}`.
- You could also highlight the difference, in the table, between the weakly supervised and the fully supervised methods, so that an inattentive reader won't believe that Unet is doing much better than you : )

**Final Rating Justification:**

I am happy with the responses that the authors provided during the rebuttal (especially when highlighting the difference with GMIC).

It seems that they promised to incorporate a lot of things into the main paper (I wish them luck to fit the page limit), but I am not overly concerned that they will manage it, as a lot of the text could be

**Justification Of The Preliminary Rating:**

I am quite conservative here, but the score could easily be upgraded to a strong accept during the rebuttal. To me, it is currently a clear accept. I will follow closely the discussion with the authors and the other reviewers.

**Paper Type:**

validation/application paper

**Questions To Address In The Rebuttal:**

- Do you think there would there be a way to share some of the weights/encoding layers of the global and local modules ? As the networks are not pruned, I assume there is quite a bit of redundancy/unused capacity in the trained networks first layers.
- Appendix A and Algorithm 1 clarity could be a bit improved. If I understood correctly, it will select only $M$ ($K$ at inference) _positive_ patches, is that correct ? What happens when an image has few tumors, and henceforth less than $M$ patches to fill with tumors ?
- It seems that the saliency sparsity loss $S_n$ is quite important. How sensitive is the method to the choice of $\gamma_0$, $\gamma_1$ and $\gamma_2$ ?
- Would adding a significant numbers of negative scans (no tumor at all) boost performances of your method, or would that imbalance the training too much ?

**Special Issue:**

yes

---

> ### Author Response · Authors · 2021-03-17
> **Response to AnonReviewer1(Part One)**
>
> We thank the reviewer for the precious comments.
> > I think the paper would benefit from mentioning that the current task is a case of multiple instance learning [1]. Not because its solutions are relevant here (it's not), but because it gives I think an interesting and relevant framing to tackle the task. This could be especially relevant in a future extension when trying to increase the resolution further. (For instance, by treating the patches selected after the first stage as positive or negative "bags", it could be plugged into a regular network utilizing such information; for instance by using a prior on how much the positive bags are filled with tumor.)
>
> We agree with the reviewer that this is relevant and it provides an interesting perspective useful to future extensions. We will add a description of the connection with multiple instance learning in the final version.
>
> > I think a discussion on the training and inference cost is missing, especially as two full networks are involved. A higher cost will absolutely not invalidate the authors method, but it would give some perspective.
>
> Thank you for the suggestion. We will include this information in the final version.
>
> > The section on the aggregation function, to train the local module, should be clearer and more prominent in the paper, as it is, I believe, the main ingredient to the method's success. Currently it is scattered across the paper when it should be one dedicated subsection in the main-matter.
>
> We agree that this part is important and will revise the final version of the manuscript accordingly.
>
> > The paper is a big long, compared to the MIDL limit of 8 pages, and a lot of the main content ends up in the appendix. [...] Nevertheless, the camera ready version could probably benefit from some reorganization between main-matter and appendix.
>
> This is another good point. We will reorganize the main paper and the appendix to make the flow smoother.
> > Your tables (such as table 1) could probably be made a tiny bit more readable, by using the features of the package booktabs. The table columns could be swapped from llllllllll to lcccccccc and the second \hline replaced by \cmidrule(lr){2-3}\cmidrule(lr){4-5}\cmidrule(lr){6-7}\cmidrule(lr){8-9}.
>
> Thanks for this useful tip! We will incorporate that in the updated version.
>
> > You could also highlight the difference, in the table, between the weakly supervised and the fully supervised methods, so that an inattentive reader won't believe that Unet is doing much better than you : )
>
> Thanks for this piece of advice. We will highlight the difference.

---

> ### Author Response · Authors · 2021-03-17
> **Response to AnonReviewer1(Part Two)**
>
> > Do you think there would there be a way to share some of the weights/encoding layers of the global and local modules ? As the networks are not pruned, I assume there is quite a bit of redundancy/unused capacity in the trained networks first layers.
>
> This is a very intriguing point! In our current design, the global network aggressively downsamples the image and has fewer channels to save computational resources. The local module, on the other hand, uses a small stride to preserve the resolution and has more channels. It may therefore be somewhat challenging to share weights, but it may still be possible to do so in some layers. This is one of the directions that we want to explore in a follow-up paper.
>
> > Appendix A and Algorithm 1 clarity could be a bit improved. If I understood correctly, it will select only M (K at inference) positive patches, is that correct ? What happens when an image has few tumors, and henceforth less than patches to fill with tumors ?
>
> Thank you for pointing out that this is unclear. We will edit the paper to fix this. We do not assume that all “positive” patches contain tumors. Only some of the M (K at inference) patches extracted by the global module will contain tumors. This is accounted for by the patch aggregation function in the local module, which produces a single classification output for the M aggregated patches. Therefore, it is OK if the image has fewer tumors than patches. However, when none of the “positive” patches contains a tumor, a standard multiple instance learning assumption that a positively labeled bag contains at least one positively labeled object is violated. This is why it is beneficial to use a larger number of patches during training (see Appendix H.4). We find that this does not occur often though. We have done an additional analysis to investigate this. We used a set of images with tumors from the validation set, and checked the first three patches extracted according to the global saliency map. These patches did not contain any lesions only in 40 out of 300 benign cases (13%) and 6 out of 75 malignant cases (8%). We will add this analysis to the paper. During inference, when an image has fewer tumors than the selected patches, we have verified that the local saliency maps are robust to the choice of K to some extent, but there is a trade-off between precision and recall (see Appendix H.5 and Figure 8).
>
> > It seems that the saliency sparsity loss is quite important. How sensitive is the method to the choice of $\gamma_0, \gamma_1, \gamma_2$?
>
> The sparsity loss is indeed important because lesions tend to be small, and promoting sparsity in the saliency map prevents highlighting irrelevant areas, as shown in previous works (e.g. [1]). We will emphasize this in the final version. $\gamma_0, \gamma_1, \gamma_2$ are not related to the sparsity loss though. They weigh the contributions of the three saliency maps in the global module as explained in Appendix B. The results are not sensitive to this choice. In fact just setting $\gamma_1$ to one yields pretty good results (see Appendix H.1). We will add an ablation study to evaluate the robustness of the results with respect to the sparsity parameter $\lambda$ in the final version of the paper.
>
> > Would adding a significant numbers of negative scans (no tumor at all) boost performances of your method, or would that imbalance the training too much ?
>
> This is an important point. Our dataset (458,852 breasts) contains more exams without lesions (452,311 compared to 5,556 with benign lesions, and 985 with malignant lesions). To account for this imbalance when training our models, at each epoch we use all exams with lesions and an equal number of randomly-sampled exams without lesions. This is explained in Appendix G, but it is an important point, so we will incorporate it in the main paper.
>
> [1] Shen, Y., Wu, N., Phang, J., Park, J., Liu, K., Tyagi, S., Heacock, L., Kim, S.G., Moy, L., Cho, K. and Geras, K.J. An interpretable classifier for high-resolution breast cancer screening images utilizing weakly supervised localization. Medical Image Analysis, 2021.

---

### Author Response · Authors · 2021-03-17
**Comment for all reviewers**

We thank the reviewers for their time and their feedback. We found them engaging and insightful. They will improve the paper in its final version. We address the specific questions and comments below.

---

### Meta-Review · Area_Chair1 · 2021-03-27

**Recommendation:** Accept (Poster)

**Metareview:**

1 SA, 2 WA, 1WR. The WR reviewer found the paper to have low contribution compared to previous work and low segmentation accuracy. All reviewers agreed that the paper has merit, it has been evaluated in a large dataset and the proposed method is supported with a lot of ablation studies. The authors addressed the points raised by the reviewers and clarified their differences with the previous work during the discussion. On balance, I agree with the reviewers that the paper has merit, it deals with a very interesting topic and I think that it will be a good contribution for MIDL 2021. During preparing the final version, the authors should address all the points raised by the reviewers.

**Paper Type:**

both

---

### Decision · Program_Chairs · 2021-03-31

Accept